# Probiotic *Bacillus* Strains Enhance T Cell Responses in Chicken

**DOI:** 10.3390/microorganisms11020269

**Published:** 2023-01-19

**Authors:** Filip Larsberg, Maximilian Sprechert, Deike Hesse, Gunnar Loh, Gudrun A. Brockmann, Susanne Kreuzer-Redmer

**Affiliations:** 1Albrecht Daniel Thaer-Institute, Breeding Biology and Molecular Genetics, Humboldt Universität zu Berlin, Unter den Linden 6, 10099 Berlin, Germany; 2Evonik Operations GmbH-Research, Development & Innovation Nutrition & Care, Kantstraße 2, 33790 Halle, Germany; 3Institute of Animal Nutrition, Nutrigenomics, University of Veterinary Medicine Vienna, Veterinärplatz 1, 1210 Vienna, Austria

**Keywords:** antimicrobial resistance, immune-modulating feed additives, probiotics, chicken, *Bacillus* spp.

## Abstract

Banning antibiotic growth promotors and other antimicrobials in poultry production due to the increasing antimicrobial resistance leads to increased feeding of potential alternatives such as probiotics. However, the modes of action of those feed additives are not entirely understood. They could act even with a direct effect on the immune system. A previously established animal-related in vitro system using primary cultured peripheral blood mononuclear cells (PBMCs) was applied to investigate the effects of immune-modulating feed additives. Here, the immunomodulation of different preparations of two probiotic *Bacillus* strains, *B. subtilis* DSM 32315 (BS), and *B. amyloliquefaciens* CECT 5940 (BA) was evaluated. The count of T-helper cells and activated T-helper cells increased after treatment in a ratio of 1:3 (PBMCs: *Bacillus*) with vital BS (CD4+: *p* < 0.05; CD4+CD25+: *p* < 0.01). Furthermore, vital BS enhanced the proliferation and activation of cytotoxic T cells (CD8+: *p* < 0.05; CD8+CD25+: *p* < 0.05). Cell-free probiotic culture supernatants of BS increased the count of activated T-helper cells (CD4+CD25+: *p* < 0.1). UV-inactivated BS increased the proportion of cytotoxic T cells significantly (CD8+: *p* < 0.01). Our results point towards a possible involvement of secreted factors of BS in T-helper cell activation and proliferation, whereas it stimulates cytotoxic T cells presumably through surface contact. We could not observe any effect on B cells after treatment with different preparations of BS. After treatment with vital BA in a ratio of 1:3 (PBMCs:*Bacillus*), the count of T-helper cells and activated T-helper cells increased (CD4+: *p* < 0.01; CD4+CD25+: *p* < 0.05). Cell-free probiotic culture supernatants of BA as well as UV-inactivated BA had no effect on T cell proliferation and activation. Furthermore, we found no effect of BA preparations on B cells. Overall, we demonstrate that the two different *Bacillus* strains enhanced T cell activation and proliferation, which points towards an immune-modulating effect of both strains on chicken immune cells in vitro. Therefore, we suggest that administering these probiotics can improve the cellular adaptive immune defense in chickens, thereby enabling the prevention and reduction of antimicrobials in chicken farming.

## 1. Introduction

Poultry is the cheapest source of animal protein worldwide and the most common source of meat. In 2030, poultry meat is expected to represent about 41% of all the protein from meat sources [1]. Furthermore, increased poultry production in developing countries [2] is accompanied by an elevated overall usage of antimicrobials. However, there is growing concern about the resistance of pathogenic bacteria against antibiotics and other antimicrobials, the residual effects of antibiotics in meat products [3], and the public health risk from zoonotic pathogens such as *Salmonella* and *Campylobacter*. Consequently, the subtherapeutic usage of antibiotic growth promotors and other antimicrobials has been forbidden in farming in the EU since 2006 (Regulation (EC) No 1831/2003). In the last decade, alternatives such as prebiotics [4,5], probiotics [4,5], synbiotics [6], and organic acids [7] have arisen to improve the health of animals and to reduce the usage of antibiotics and other antimicrobials. The most promising feed additives comprise probiotics, representing mainly live bacteria, fungi, and yeast, which supplement the gastrointestinal flora and promote the maintenance of a healthy digestive system. Probiotic usage has increased over the last years due to antibiotic-free poultry production and its well-researched benefits [8]. Each probiotic strain possesses varying levels of beneficial effects for the host [8]. These include the secretion of antimicrobial substances in the gastrointestinal tract (GIT), competitive adherence to the gut mucosa. the epithelium, as well as the improvement of the gut epithelial barrier and its integrity [9]. Moreover, probiotics improve growth performance [10], nutrient digestibility [11,12], and modulate the immune system of the host [13,14,15] as the interaction between host cells and the bacteria or their structural components may lead to modulation of either local or systemic T lymphocyte- (T cell) and B lymphocyte (B cell)-mediated immune responses [16]. Among the most widely and commonly used in-feed probiotics are *Bacillus* spp., which are spore-forming bacteria with resistance to high temperatures and harsh storing conditions [17]. These are generally considered as safe strains for use in poultry production [18,19] and comprise different positive effects on chicken health. Thus, the commercially available probiotic strains *Bacillus amyloliquefaciens* CECT 5940 (BA) and *Bacillus subtilis* DSM 32315 (BS) were reported to be effective in reducing or eliminating the negative effects caused by pathogenic bacteria such as *Escherichia coli* and *Clostridium perfringens* [20,21,22,23,24,25]. Moreover, they were shown to improve growth performance, increase feed efficiency, and manipulate the intestinal structure and microbial composition of the gut [20,24,26]. Furthermore, a positive effect on the immune system was reported in vivo [26,27]. An intact immune system in the GIT, the main portal of entry for pathogenic microorganisms, is indispensable. The two major cell types of the adaptive immune system are T and B cells. B cells contribute to the adaptive immune response through the secretion of antibodies, known as the humoral immune response, and can be recognized with an antibody against the chB6 (Bu-1) marker, a highly glycosylated protein of unique structure in chickens [28]. T cells mainly act as the cellular component of the adaptive immune response. The chicken T cell receptor (TCR) can be assigned to three subgroups which are identified by specific monoclonal antibodies [29]. γδ T cells, encompassing up to 50% of the peripheral blood T cells [30], can be recognized by the TCR1 antibody, whereas the two discrete subsets of αβ T cells can be labeled and recognized by TCR2 and TCR3 antibodies [29]. Like in humans and mice, αβ T cells can be further distinguished and subdivided by the expression of their co-receptors, cluster of differentiation 4 (CD4) and CD8 representing CD4+ T-helper cells and CD8+ cytotoxic T cells [31,32]. T and B cells exhibit distinct functions in mediating the immune response and are known to confer immunological protection by developing pathogen-specific memory [33].

This study aimed to assess, if the two probiotic *Bacillus* strains BA and BS are able to directly modulate two major components of the adaptive immune response, T and B cells, in chicken. A recently reported in vitro chicken immune cell assay utilizing peripheral blood mononuclear cells (PBMCs) [34] was used to investigate possible effects on the immune system. 

By increasing the understanding of the functional mechanisms of probiotics on immune cells, this study can contribute to strategizing prevention measures and thereby help the poultry industry in raising healthy chickens. In particular, it will help to prevent and treat diseases by targeted administration of probiotics and thus prevent and reduce the usage of the subtherapeutic administration of antimicrobials.

## 2. Materials and Methods

### 2.1. Animals, Housing, Feeding, and Tissue Collection

The study was approved by the local state office of occupational health and technical safety “Landesamt für Gesundheit und Soziales Berlin” (LaGeSo Reg. T 0151/19). In total, 55 four-to-six-week-old broiler chickens of the commercial layer Cobb500 (Cobb Germany Avimex GmbH, Wiedemar, Germany) were used. The birds were kept and fed as described previously [34]. In brief, all chickens were fed a starter diet from day 1 to day 14 post hatch and a grower diet afterwards (H. Wilhelm Schaumann GmbH, Pinneberg, Germany). The ration was fed on an *ad libitum* basis and water was always available. The chickens were kept in groups of approximately 20 broiler chickens in 4 m² floor pens. For experiments, 3 to 6 birds per week were stunned and blood was sampled in sodium citrate (Na-citrate) pre-filled polystyrene tubes (VACUETTE^®^, Greiner Bio-One, Kremsmünster, Austria) by neck cutting.

### 2.2. PBMC Isolation and Culture

PBMCs were isolated using combined dextran–ficoll isolation as described previously [34,35]. In brief, blood was diluted with Dulbecco’s phosphate-buffered saline (DPBS, Gibco™, ThermoFisher Scientific, Waltham, MA, USA) containing 2 mM ethylenediaminetetraacetic acid (EDTA, Carl Roth, Karlsruhe, Germany) 1:1, mixed with 3% dextran (Carl Roth) in a ratio of 1:0.4, and centrifuged at 50× *g* for 20 min. Thereafter, the upper phase was collected, layered onto an equal volume of Histopaque^®^-1077 (Sigma-Aldrich, St. Louis, MO, USA), and centrifuged at 900× *g* for 30 min. The PBMCs at the interphase were collected, washed twice with DPBS/EDTA in a new 50 mL tube, and centrifuged at 400× *g* for 10 min. Isolated PBMCs were adjusted to 5×10^6^ cells/mL living cells using Tali™ image-based cytometer (ThermoFisher Scientific) with propidium iodide (PI, ThermoFisher Scientific) as a viability marker. PBMCs were cultured in RPMI 1640 medium (Gibco™, ThermoFisher Scientific) with 10% chicken serum (Gibco™, ThermoFisher Scientific) and 1% penicillin (10,000 U/mL)-streptomycin (10,000 µg/mL) (Pen/Strep, Gibco™, ThermoFisher Scientific) at 41 °C with 5% CO_2_. The next day, further experiments were performed.

### 2.3. Bacterial Strains and Culture

RPMI 1640 medium was inoculated with either *B. amyloliquefaciens* CECT 5940 (BA) or *B. subtilis* DSM 32315 (BS) and bacteria were cultured overnight at 37 °C and 120 rpm. To define the actual colony forming units per milliliter (cfu/mL), the bacterial growth was measured on a Tecan Infinite^®^ M200 Pro plate reader with Magellan™ v. 7.1 software (Tecan Group AG, Männedorf, Switzerland) and bacterial cultures were plated on tryptic-soy agar (TSA, Carl Roth) on Petri dishes (Greiner Bio-One, Frickenhausen, Germany) and counted the next day.

### 2.4. Probiotic Treatment/Co-culture

Co-culture experiments with probiotic bacterial strains were performed with 1×10^6^ PBMCs with different preparations of either BS or BA in a total volume of 1 mL in 24-well plates (Eppendorf, Hamburg, Germany) for 24 h. All experiments were performed in RPMI 1640 medium containing 10% chicken serum without Pen/Strep at 41 °C with 5% CO_2_. For all co-culture experiments, concanavalin A (conA, Vector Laboratories, Newark, California, USA) was used as a positive control. PBMCs were treated at a concentration of 10 µg/mL conA (Appendix A).

#### 2.4.1. Treatment with Vital Probiotic Bacteria

To investigate the effect of vital probiotic bacteria, 1×10^6^ PBMCs were co-cultured with either BS or BA in a ratio of 1:3 (PBMCs:*Bacillus*).

#### 2.4.2. Treatment with Supernatants of Probiotic Culture

For treatment with cell-free supernatants of the probiotic bacterial cultures, 40 mL RPMI 1640 medium was inoculated with either BS or BA, and bacteria were cultured overnight at 37 °C and 100 rpm. The next day, the bacterial cultures were sterile-filtered using a vacuum filter system with a polyethersulfone membrane possessing a pore size of 0.22 μm (Corning, Corning, NY, USA). The supernatants of the cultures were then split into two new 50 mL tubes. For thermal inactivation of the secreted factors of the bacterial strains, one 50 mL tube was heat-treated (10 min at 80 °C), and the other 50 mL tube was left untreated. Heat-treated and untreated pure RPMI 1640 medium served as a negative control. 1×10^6^ PBMCs were exposed to either 500 µL heat-treated supernatants, untreated supernatants, heat-treated RPMI 1640 medium, or untreated RPMI 1640 medium. Treatment occurred in a total volume of 1 mL for 24 h. All experiments with cell-free bacterial culture supernatants were performed in RPMI 1640 medium containing 10% chicken serum and without Pen/Strep at 41 °C with 5% CO_2_. To validate that cell-free supernatants were used, bacterial culture supernatants were plated on TSA agar plates before and after sterile filtering.

#### 2.4.3. Treatment with UV-inactivated Probiotic Bacteria

To investigate whether secreted factors or cell surface proteins could influence immune cell phenotypes, the bacteria were inactivated by UV light. Therefore, 20 mL RPMI 1640 medium was inoculated with either BS or BA and cultured overnight at 37 °C. On the next day, bacterial cultures were transferred into Petri dishes and irradiated with UV light (200 nm) for 5 h. Thereafter, the bacteria were harvested and stored at 4 °C overnight for co-culture experiments. The next day, 1×10^6^ PBMCs were treated with either UV-inactivated BS or BA in a ratio of 2:1 (PBMCs:*Bacillus*) in a total volume of 1 mL for 24 h. The number of bacteria used for treatment was determined before the UV inactivation. To validate the UV inactivation, bacterial cultures were plated on TSA agar plates before and after UV light exposure.

### 2.5. Immunophenotyping

For immunophenotyping using flow cytometry, cells were harvested, centrifuged for 10 min at 400× *g*, washed once with DPBS/EDTA containing 0.05% bovine serum albumin (BSA, Sigma-Aldrich), and stained with different panels of monoclonal antibodies. In this study, CD4+ T cells were stained with mouse anti-chicken CD4-phycoerythrin (PE) or CD4-SpectralRed (SPRD) (CT-4, SouthernBiotech, Birmingham, Alabama, USA). CD8+ T cells were labeled with CD8-allophycocyanin (APC) (CT-8, ThermoFisher Scientific). In addition, human anti-chicken CD25-FITC (AbD13504, Bio-Rad Laboratories, Hercules, California, USA) and CD28-PE (AV7, SouthernBiotech) were used to investigate the activation of CD4+ T-helper cells (CD4+CD25+, CD4+CD28+) and CD8+ cytotoxic T cells (CD8+CD25+, CD8+CD28+). All CD3+ T cells were labeled with the pan-T-cell marker CD3-APC (CT-3, ThermoFisher Scientific). Bu-1+ B cells were stained with Bu-1-FITC (AV20, ThermoFisher Scientific). As a viability marker, 1.5 µL of 1 mg/mL 4′, 6-diamino-2-phenylindole (DAPI, ThermoFisher Scientific) was used. 20,000 vital PBMCs were recorded on a BD FACSCanto™ II (Becton, Dickinson and Company, Franklin Lakes, NJ, USA) flow cytometer.

### 2.6. Statistical Analysis

The data obtained by flow cytometry was analyzed using a gating strategy (Appendix A). The cell count of antibody-positive cells was calculated relative to the vital PBMCs measured on a BD FACSCanto™ II. Due to high variation in the immune response capacity, the results are displayed as the ∆ relative cell count, representing the difference of the relative cell count between the treatment and the respective untreated negative control for every biological replicate. Statistical analysis was performed by a one sample Student’s *t*-test. All tests were performed using GraphPad Prism 8.0.2 (GraphPad Software, San Diego, CA, USA). Differences between groups were considered statistically significant at *p* < 0.05 (* = *p* < 0.05, ** = *p* < 0.01, *** = *p* < 0.001, **** = *p* < 0.0001). A statistical tendency is shown as +, *p* < 0.1.

## 3. Results

### 3.1. Effects of Vital BS and BA on T and B Cells in a PBMC Composite

The effect of vital BS and BA on chicken adaptive immune cells was evaluated by measures of the proportions of the adaptive immune cell populations. In particular, the proliferation of CD4+ T-helper cells (Figure 1a), CD8+ cytotoxic T cells (Figure 1b), and all CD3+ T cells (Figure 1c) were measured. Furthermore, the effect on the proliferation of Bu1+ B cells was investigated (Figure 1d). Additional T cell activation markers (CD25, CD28) allowed to display the effect on CD4+CD25+ activated T-helper cells (Figure 2a), CD8+CD25+ activated cytotoxic T cells (Figure 2b), and all CD28+ αβ T cells (Figure 2c).

The treatment of PBMCs with vital BS revealed an increase of the ∆ relative cell count of CD4+ T-helper cells by 4.10% (*p* < 0.05, Figure 1a) and CD8+ cytotoxic T cells by 1.79% (*p* < 0.05, Figure 1b). Furthermore, the ∆ relative cell count of all CD3+ T cells increased by 7.83% (Figure 1c). However, this result was not significant. Additionally, treatment with vital BS had no effect on Bu-1+ B cells (Figure 1d). 

After treatment with vital BA, the ∆ relative cell count of CD4+ T-helper cells increased by 2.38% (*p* < 0.1, Figure 1a), while the count of CD8+ cytotoxic T cells (Figure 1b) remained unaffected. The ∆ relative cell count of all CD3+ T cells increased by 2.64% (Figure 1c), however, the result was not significant. The ∆ relative cell count of Bu-1+ B cells did not change after treatment with vital BA (Figure 1d).

Corresponding to the increase of the ∆ relative cell count of CD4+ T-helper cells and CD8+ cytotoxic T cells, vital BS enhanced the count of CD4+CD25+ activated T-helper cells by 0.67% (*p* < 0.01, Figure 2a) and CD8+CD25+ activated cytotoxic T cells (*p* < 0.05, Figure 2b) by 0.16%. Additionally, activated CD28+ αβ T cells increased by 5.20% (*p* < 0.01, Figure 2c) after treatment with BS.

The addition of vital BA resulted in an elevated number of the CD4+CD25+ activated T-helper cell proportion by 0.37% (*p* < 0.05, Figure 2a). The ∆ relative cell count of CD8+CD25+ activated cytotoxic T cells (Figure 2b) remained unaffected after treatment with vital BA, while all CD28+ αβ T cells increased by 3.81% (*p* < 0.05, Figure 2c).

Summarized, vital BS stimulated the proliferation and activation of T-helper cells (CD4+, CD4+CD25+), cytotoxic T cells (CD8+, CD8+CD25+) and all αβ T cells (CD28+), while vital BA stimulated the activation and proliferation of T-helper cells (CD4+, CD4+CD25+) and all αβ T cells (CD28+).

### 3.2. Effects of Cell-Free Culture Supernatants of BS and BA on T and B cells in a PBMC Composite

After treatment with vital probiotic bacteria, we evaluated which component of the probiotics BS and BA is involved in the stimulation of the immune system, especially T cells. Therefore, in a first step, we analyzed the effects of cell-free bacterial culture supernatants of BS and BA on PBMCs by measuring the proliferation (Figure 3a–d) and the activation status (Figure 4a,b) of the adaptive immune cell populations, T and B cells.

PBMC treatment with cell-free culture supernatants of the probiotic BS did not affect the CD4+ T-helper cells (Figure 3a) and CD8+ cytotoxic T cells (Figure 3b). Furthermore, the ∆ relative cell count of all CD3+ T cells (Figure 3c) and Bu-1+ B cells (Figure 3d) remained unaffected after treatment. Interestingly, we found a decreased number of CD4+ T-helper cells after treatment with the negative control, heat-treated cell-free culture supernatants of BS (Appendix A).

After treatment of PBMCs with cell-free culture supernatants of BA, we found no effect on the Δ relative cell count of CD4+ T-helper cells (Figure 3a), CD8+ cytotoxic T cells (Figure 3b), all CD3+ T cells (Figure 3c), and Bu-1+ B cells (Figure 3d). In addition, we found a decreased count of Bu-1+ B cells after exposure to heat-treated cell-free culture supernatants of BA (Appendix A).

For CD4+CD25+ activated T-helper cells, the ∆ relative cell count was tendentially increased by 0.39% (*p* < 0.1, Figure 4a) after treatment with cell-free culture supernatants of BS, whereas CD8+CD25+ activated cytotoxic T cells (Figure 4b) were not affected.

After treatment of PBMCs with cell-free culture supernatants of BA, we found no effect on the Δ relative cell count of CD4+CD25+ activated T-helper cells (Figure 3a) and CD8+CD25+ activated cytotoxic T cells (Figure 3c,d).

Summarized, secreted factors, expected in the cell-free culture supernatants of BS, stimulated T-helper cell activation (CD4+CD25+). Furthermore, heat-treated cell-free culture supernatants, which are supposed to contain denaturated proteins, decreased the CD4+ T-helper cell count. Cell-free culture supernatants of BA had no effect on T and B cells. However, heat-treated cell-free culture supernatants decreased the count of Bu-1+ B cells.

### 3.3. Effects of UV-Inactivated BS and BA on T and B Cells

As we could partially confirm the involvement of secreted factors in the stimulation of T-helper cells by BS and could further exclude the involvement of secreted factors in the stimulation of cytotoxic T cells by BS and of T-helper cells by BA, we investigated the effect of the bacterial cell surface. We hypothesized that direct contact of the bacterial cell surface and the adaptive immune cells constitutes the major component of adaptive immune cell activation. To this end, the bacteria were inactivated by UV light. After UV light exposure, we assumed that the bacteria possessed an intact cell surface, as reported earlier [36], which could be involved in a possible immunomodulatory effect. The proliferation (Figure 5a–d) and the activation status (Figure 6a,b) of the adaptive immune cell populations, T and B cells, were investigated.

After the treatment of PBMCs with UV-inactivated BS, we did not observe a difference in the Δ relative cell count of CD4+ T-helper cells (Figure 5a). In contrast, treatment with UV-inactivated BS significantly increased the CD8+ cytotoxic T cell count by 2.96% (*p* < 0.01, Figure 5b). Furthermore, the CD3+ T cell population was enhanced after treatment with UV-inactivated BS by 2.61% (Figure 5c), although, not significantly. The Bu-1+ B cell count remained unaffected after treatment with UV-inactivated bacteria (Figure 5d).

Treatment with UV-inactivated BA did not result in a change of the Δ relative cell count of CD4+ T-helper cells (Figure 5a), CD8+ cytotoxic T cells (Figure 5b), the CD3+ T cell population (Figure 5c), and Bu-1+ B cells (Figure 5d).

We did not observe an effect on CD4+CD25+ activated T-helper cells (Figure 6a) after treatment with UV-inactivated BS. In contrast, the ∆ relative cell count of the CD8+CD25+ activated cytotoxic T cell population seemed to be increased by 0.19% (Figure 6b) after treatment with UV-inactivated BS, however, the result was not significant.

Treatment with UV-inactivated BA did not result in a change of the Δ relative cell count of CD4+CD25+ activated T-helper cells (Figure 6a) and CD8+CD25+ activated cytotoxic T cells (Figure 6b).

In summary, we observed that UV-inactivated BS increased the count of CD8+ cytotoxic T cells and seemed to stimulate cytotoxic T cell activation (CD8+CD25+). In contrast, UV-inactivated BA had no effect on CD4+ T-helper cell, CD8+ cytotoxic T cell, and Bu-1+ B cell responses in PBMCs of broiler chicken.

## 4. Discussion

The use of antimicrobials, especially antibiotics, in broiler production triggers the development of antimicrobial-resistant pathogenic bacteria with implications for human and animal health. Innovative feeding strategies, including the use of probiotics, have the potential to support animal health and, thus, may help to reduce the use of antimicrobials in poultry production. Proposed modes of probiotic action include immunomodulatory effects of specific strains. Primary cell culture systems represent a good alternative to animal experiments, more closely mimicking the in vivo model compared to conventional cell lines, to investigate the mechanisms of potentially immune-modulating feed additives. In this study, we investigated the immunomodulatory potential of two probiotic *Bacillus* strains, BS and BA, which are both commercially available probiotics for chicken farming, using an in vitro cell culture system with chicken immune cells [34].

We detected stimulating effects of both probiotic *Bacillus* strains on the proliferation (percentage of CD4+ T-helper cells) (Figure 1a), as well as on the activation (percentage of CD4+CD25+ activated T-helper cells) (Figure 2a) of T-helper cells. Furthermore, the count of activated CD28+ αβ T cells (Figure 2c) increased after treatment with both bacterial strains, which is in line with the increase of the CD4+ and the CD4+CD25+ activated T-helper cell proportion (Figure 1a and Figure 2a). CD28, a costimulatory molecule important for T cell activation [37], is reported to be a late activation marker (48 h) and only for CD8+ cytotoxic T cells [38]. In our study, the CD28 signal increased significantly 24 h after treatment with BS and BA. These results hint towards an in vitro stimulation of CD4+ and CD4+CD25+ activated T-helper cells through vital BS and BA bacteria. CD4+CD25+ T cells are further reported to possess properties similar to that of mammalian regulatory T cells (Tregs) [39,40]. Therefore, chicken Tregs are required to maintain immune homeostasis and self-antigen tolerance. Interestingly, the CD4+CD25+ T cell population showed no downregulation of the cytotoxic T cell activity in our study and therefore did not seem to possess suppressing properties, as it was reported earlier [41]. The proliferation (percentage of CD8+ T cells) and activation (percentage of CD8+CD25+ T cells) of cytotoxic T cells increased after treatment with vital BS (Figure 1b and Figure 2b). In another study, the applicability of chicken homologs to well-known mammalian T cell activation markers was investigated [38]. Next to CD28, CD5, MHC-II, CD44, and CD45, the marker CD25 was used to measure T cell activation. The frequencies of CD25+ T cells were increased after conA stimulation for 24 h, which demonstrated that CD25 could also be used for chicken PBMCs as an activation marker [38]. Therefore, we used CD25 as a T cell activation marker. In addition to the effect of vital bacteria, the activation of T-helper cells tended to increase after treatment with cell-free culture supernatants of BS (CD4+CD25+, Figure 4b). This result points towards an involvement of secreted factors of the bacteria, which remained in the supernatants, in the proliferation (CD4+) and activation (CD4+CD25+) of T-helper cells. Interestingly, we found a decrease in the CD4+ T-helper cell count after treatment with heat-treated cell-free culture supernatants of BS (Appendix A). This result possibly indicates important compounds for CD4+ T-helper cell survival, which were inactivated by heat in the cell-free bacterial culture supernatants. In the experiments with vital BS, we found stimulating effects on the proliferation (CD4+) and activation (CD4+CD25+) of T-helper cells (Figure 1a and Figure 2a) as well as on the proliferation (CD8+) and activation (CD8+CD25+) of cytotoxic T cells (Figure 1b and Figure 2b). This finding is in line with observations in a previous in vivo study in chicken with probiotic *Lactobacillus acidophilus* LA5 [42]. Moreover, *B. subtilis* spores, as adjuvants, were shown to enhance CD4+ and CD8+ T cell responses in vivo against avian influenza H9N2 [43]. In a necrotic enteritis challenge, *B. subtilis* DSM 32315, the same bacterial strain used in our study, was demonstrated to reduce pathology and improve the performance of broilers [23]. We show that UV-inactivated BS bacteria led to an increase of CD8+ cytotoxic T cells (Figure 5b), whereas the median after treatment with cell-free supernatants of both bacterial strains rather tended to decrease (Figure 3b). This result indicates an involvement of direct contact of the cell surface of BS with the immune cells for cytotoxic T cell activation. UV irradiation can denature the DNA of microorganisms, causing death or inactivation [44], however, the cell surface of UV-inactivated bacteria was reported to remain intact [36]. Additionally, the result obtained for all CD3+ T cells after treatment with UV-inactivated BS (Figure 5c) points to the same direction regarding an effect on the T cell population. However, probably partially due to a smaller sample size in this experiment, those results were not significant. In mice, *B. subtilis* was shown to induce anti-inflammatory M2 macrophages inhibiting T cell-mediated immune responses [45]. From our results, we suggest that BS may improve the performance of the animals against infection. Thus, BS helps to prevent and treat disease through the modulation of T cell responses in chicken.

B cells, another important cell type of the adaptive immune response, play a major role in the antibody-mediated humoral immune response. Probiotic effects on B cells have mainly focused on antibody production, like immunoglobulin A (IgA) production in the GIT [33,46,47]. Moreover, probiotics were reported to affect B cells by altering or increasing the number of antigen-specific B cells, as it was shown after in vitro stimulation of human PBMCs with different lactic acid bacterial (LAB) strains [48] and in chicken after vaccination with inactivated avian influenza H9N2 together with a *B. subtilis* spore adjuvant [43]. In pigs, B cell stimulation in PBMCs was also reported after in vivo administration of probiotic *Enterococcus faecium* NCIMB 10415 in piglets [49]. For Bu-1+ B cells, we found no effect of both vital probiotic *Bacillus* strains, BS and BA (Figure 1d). Furthermore, we found no differences after treatment with cell-free supernatants (Figure 3d) and UV-inactivated bacteria (Figure 5d). However, after treatment with heat-treated cell-free culture supernatants, we found a significant decrease in the Bu-1+ B cell population (Appendix A), possibly indicating an apoptosis-inducing effect of the denatured proteins or the loss of factors necessary for B cell survival.

This study provides evidence of a direct immunomodulatory effect of BS and BA on chicken T cells. Treatment with BS increased the proliferation (CD8+) and activation (CD8+CD25+) of cytotoxic T cells, which may enhance the cellular immune response, in particular by destroying pathogens through perforins and granzymes or inducing apoptosis of infected host cells [50]. Furthermore, treatment with both probiotic bacteria, BS and BA, increased the proliferation (CD4+) and activation (CD4+CD25+) of T-helper cells. However, the CD4+CD25+ T cell population could further represent Tregs, regulating and suppressing excessive T-helper cell 1 (Th1) and T-helper cell 2 (Th2) immune responses and maintaining immunological unresponsiveness and tolerance to self-antigens [40,50,51].

## 5. Conclusions

Our results point towards a possible involvement of secreted factors of BS in T-helper cell activation and proliferation, whereas BS stimulated cytotoxic T cells presumably through surface contact. For BA, more experiments are necessary to unravel the mechanism behind the action on T cells. All in all, we observed a T cell stimulation by both tested probiotic *Bacillus* strains in vitro. Therefore, we suggest that these probiotic *Bacillus* strains could improve the health of animals and their defense against infection and, thus, help to prevent and reduce the usage of antimicrobials in chicken farming.

## Figures and Tables

**Figure 1 microorganisms-11-00269-f001:**
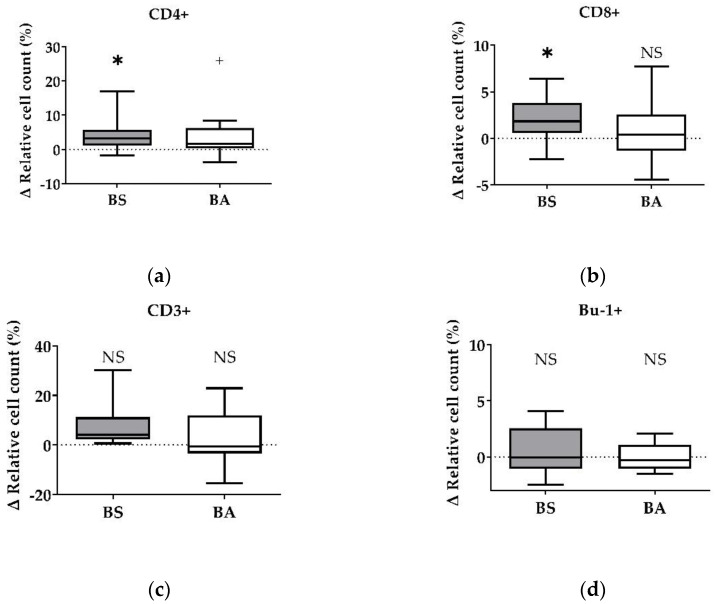
Influence of vital *B. subtilis* DSM 32315 (grey) and *B. amyloliquefaciens* CECT 5940 (white) on the proliferation of T and B cells. Isolated PBMCs were treated with BS or BA in a ratio of 1:3 (PBMCs:*Bacillus*) or left untreated (negative control). After treatment for 24 h, immune cells were stained with monoclonal antibodies. The relative cell count was related to untreated controls. DAPI was used as a viability marker. (**a**) CD4+ T-helper cells, (**b**) CD8+ cytotoxic T cells, (**c**) all CD3+ T cells, and (**d**) Bu-1+ B cells relative to the vital PBMC cell count. 20,000 vital PBMCs were recorded on a BD FACSCanto II flow cytometer. Data represent the following numbers of biological replicates: 11 (BS, BA) for CD4+, 13 (BS), and 11 (BA) for CD8+, 6 (BS), and 9 (BA) for CD3+ and Bu-1+. Results are presented as box and whisker plots showing the median, with a 25–75 percentile range as the box and minimum to maximum as the whiskers. Significance is shown as +, *p* < 0.1; *, *p* < 0.05; NS: not significant.

**Figure 2 microorganisms-11-00269-f002:**
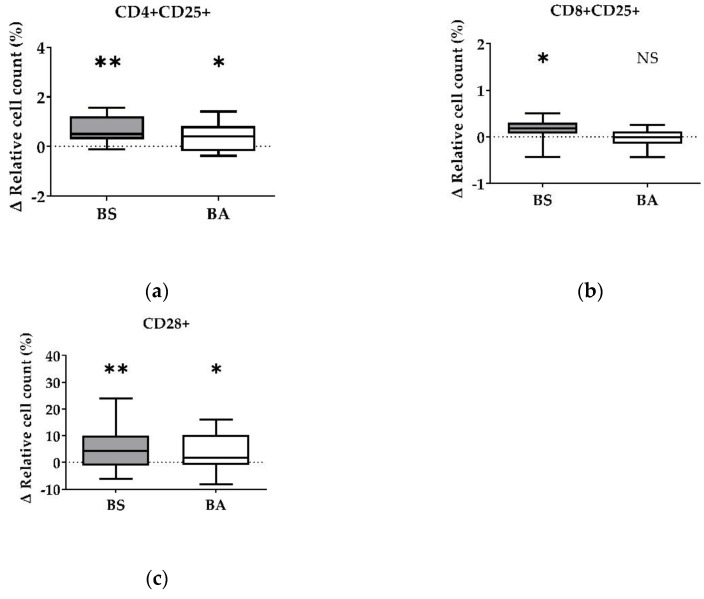
Influence of vital *B. subtilis* DSM 32315 (grey) and *B. amyloliquefaciens* CECT 5940 (white) on activation of T cells. Isolated PBMCs were treated with BS or BA in a ratio of 1:3 (PBMCs: *Bacillus*), or left untreated (negative control). After treatment for 24 h, immune cells were stained with monoclonal antibodies. The relative cell count was related to untreated controls. DAPI was used as a viability marker. (**a**) CD4+CD25+ activated T-helper cells, (**b**) CD8+CD25+ activated cytotoxic T cells, (**c**) activated CD28+ αβ T cells relative to the vital PBMC cell count. 20,000 vital PBMCs were recorded on a BD FACSCanto II flow cytometer. Data represent the following numbers of biological replicates: 11 (BS, BA) for CD4+CD25+, 13 (BS) and 11 (BA) for CD8+CD25+, and 24 (BS) and 22 (BA) for CD28+. Results are presented as box and whisker plots showing the median, with a 25–75 percentile range as the box and minimum to maximum as the whiskers. Significance is shown as *, *p* < 0.05; **, *p* < 0.01; NS: not significant.

**Figure 3 microorganisms-11-00269-f003:**
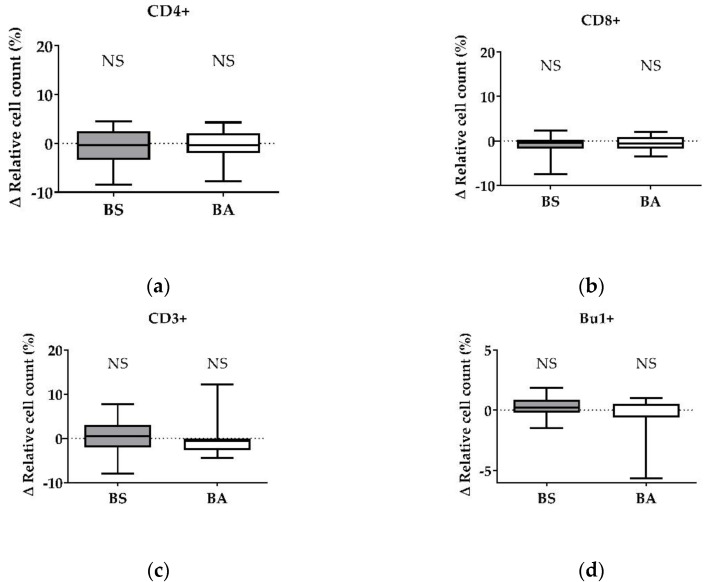
Influence of cell-free culture supernatants of *B. subtilis* DSM 32315 (grey) and *B. amyloliquefaciens* CECT 5940 (white) on the proliferation of T and B cells. Isolated PBMCs were treated with cell-free supernatants of BS or BA, or left untreated (negative control). After treatment for 24 h, immune cells were stained with monoclonal antibodies. The relative cell count was related to heat-treated controls. DAPI was used as a viability marker. (**a**) CD4+ T-helper cells, (**b**) CD8+ cytotoxic T cells, (**c**) all CD3+ T cells, and (**d**) Bu-1+ B cells relative to the vital PBMC cell count. 20,000 vital PBMCs were recorded on a BD FACSCanto II flow cytometer. Data represent 13 (BS) or 12 (BA) biological replicates. Results are presented as box and whisker plots showing the median, with a 25–75 percentile range as the box and minimum to maximum as the whiskers. Significance is shown as NS: not significant.

**Figure 4 microorganisms-11-00269-f004:**
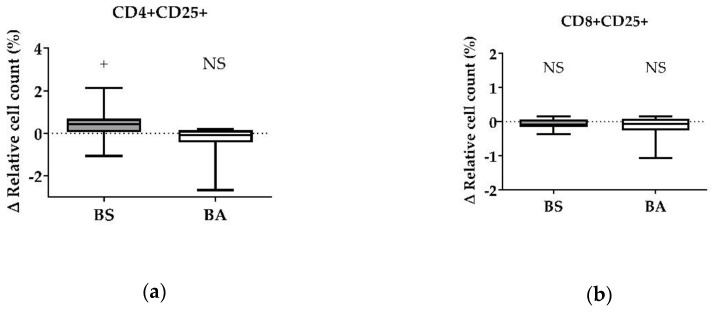
Influence of cell-free culture supernatants of *B. subtilis* DSM 32315 (grey) and *B. amyloliquefaciens* CECT 5940 (white) on the activation of T cells. Isolated PBMCs were treated with either cell-free supernatants of BS or BA, or left untreated (negative control). After treatment for 24 h, immune cells were stained with monoclonal antibodies. The relative cell count was related to heat-treated controls. DAPI was used as a viability marker. (**a**) CD4+CD25+ activated T-helper cells and (**b**) CD8+CD25+ activated cytotoxic T cells relative to the vital PBMC cell count. 20,000 vital PBMCs were recorded on a BD FACSCanto II flow cytometer. Data represent 13 (BS) or 12 (BA) biological replicates. Results are presented as box and whisker plots showing the median, with a 25–75 percentile range as the box and minimum to maximum as the whiskers. Significance is shown as +, *p* < 0.1; NS: not significant.

**Figure 5 microorganisms-11-00269-f005:**
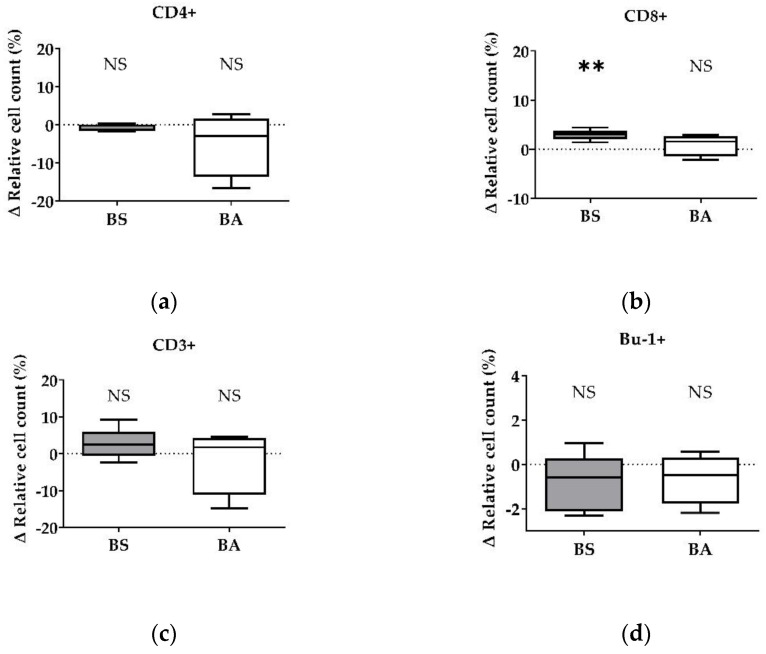
Influence of UV-inactivated *B. subtilis* DSM 32315 (grey) and *B. amyloliquefaciens* CECT 5940 (white) on the proliferation of T and B cells. Isolated PBMCs were treated with either UV-inactivated BA or BS or left untreated (negative control). After treatment for 24 h, immune cells were labeled with monoclonal antibodies. The relative cell count was related to untreated controls. DAPI was used as a viability marker. (**a**) CD4+ T-helper cells, (**b**) CD8+ cytotoxic T cells, (**c**) all CD3+ T cells, and (**d**) Bu-1+ B cells relative to the vital PBMC cell count. 20,000 vital PBMCs were recorded on a BD FACSCanto II flow cytometer. Data represent 4 (BA) or 5 (BS) biological replicates. Results are presented as box and whisker plots showing the median, with a 25–75 percentile range as the box and min to max as the whiskers. Significance is shown as **, *p* < 0.01; NS: not significant.

**Figure 6 microorganisms-11-00269-f006:**
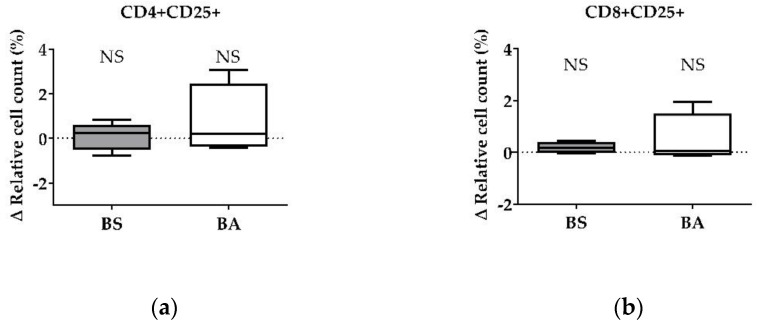
Influence of UV-inactivated *B. subtilis* DSM 32315 (grey) and *B. amyloliquefaciens* CECT 5940 (white) on the activation of T cells. Isolated PBMCs were treated with either UV-inactivated BA or BS, or left untreated (negative control). After treatment for 24 h, immune cells were stained with monoclonal antibodies. The relative cell count was related to untreated controls. DAPI was used as a viability marker. (**a**) CD4+CD25+ activated T-helper cells and (**b**) CD8+CD25+ activated cytotoxic T cells relative to the vital PBMC cell count. 20,000 vital PBMCs were recorded on a BD FACSCanto II flow cytometer. Data represent 4 (BA) or 5 (BS) biological replicates. Results are presented as box and whisker plots showing the median, with a 25–75 percentile range as the box and min to max as the whiskers. Significance is shown as NS: not significant.

## Data Availability

Data supporting reported results will be provided upon request.

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
