# Peer review of "Probiotic Bacillus Strains Enhance T Cell Responses in Chicken"

_microorganisms, 2023, doi:10.3390/microorganisms11020269_

Round 1
Reviewer 1 Report
This is an interesting study, and the manuscript is fairly well written. The manuscript has minor issues with English, however, the manuscript can be accepted following minor revisions.
Line 80-90: These sentences are repetitive. Concise and rewrite.
Line 84-86: This sentence is misleading. Correct as “In addition, the commercially available probiotic strains Bacillus amyloliquefaciens CECT 5940 (BA) [23] and Bacillus subtilis DSM 32315 (BS) [24] were reported to be effective in reducing or eliminating the negative effects caused by pathogenic bacteria such as Escherichia coli and Clostridium perfringens [23][25][26].”
Line 110-111: Correct as “By increasing the understanding of the functional mechanisms of probiotics on immune cells, this study can contribute in strategizing prevention measures and thereby help the poultry industry in raising healthy chickens”
Line 115-124: If the authors are adding details about Animals, housing, feeding, and tissue collection, add details of how many birds were used in the study; type of housing - floor pens or cages? Add details of experimental design; when and how many birds were used for sample collection?
Author Response
Dear reviewer,
please find attached the response to your comments and suggestions.
Kind regards,
Filip Larsberg

Reviewer 2 Report
The direct effects of BS and BA were evaluated on chicken PBMCs. The results demonstrated the involvement of secreted factors of BS in T-helper cell activation and proliferation. BS stimulated cytotoxic T cells by surface contact. BA treatment activated the proliferation of T-helper cells. This study indicated an immune modulating effect of Bacillus strains on chicken immune cells through activation of the T cell immune response. These findings suggest that BS and BA supplementation could contribute to preventing and reducing the usage of antimicrobials.
Author Response
Dear reviewer,
please find attached the response to your comments.
Kind regards,
Filip Larsberg
